# Removal of organic contaminants from wastewater with GO/MOFs composites

**Fuhua Wei** [ID]¹\*, **Huan Zhang**¹, **Qinhui Ren**¹, **Hongliang Chen**¹, **Lili Yang**¹, **Bo Ding**¹, **Mengjie Yu**¹, **Zhao Liang**²\*

1 College of chemistry and chemical Engineering, Anshun University, Anshun, PR China, 2 State Key Laboratory of Advanced Design and Manufacturing for Vehicle Body, College of Mechanical and Vehicle Engineering, Hunan University, Changsha City, P.R. China

\* yyspy@hnu.edu.cn, wfh.1981@163.com (FW); wallliang@163.com (ZL)

**Data Availability Statement:** All relevant data are within the manuscript and its Supporting Information files.

**Funding:** This study was funded by Guizhou Education Department Youth Science and

## Abstract

Graphene oxide/metal-organic frameworks (GO/MOFs) have been prepared via solvothermal synthesis with ferrous sulfate heptahydrate, zirconium acetate and terephthalic acid for the purpose of removing organic pollutants from wastewater. The composites were analyzed using scanning electron microscopy, infrared spectrometry, and XRD. Tetracycline hydrochloride and orange II were implemented as model pollutants to evaluate the efficacy of the GO/MOFs in water purification, in which 50 mg of Zr/Fe-MOFs/GO was mixed with 100 mL of 10 mg/L, 20 mg/L, 30 mg/L, or 50 mg/L tetracycline hydrochloride solution and 25 mg/L, 35 mg/L, 45 mg/L, or 60 mg/L orange II solution, respectively. The removal efficacy after 4 hours was determined to be 96.1%, 75.8%, 55.4%, and 30.1%, and 98.8%, 91.9%, 71.1%, and 66.2%, respectively. The kinetics of pollutant removal was investigated for both tetracycline hydrochloride and orange II and excellent correlation coefficients of greater than 0.99 were obtained. The high efficacy of these MOFs in pollutant removal, coupled with their inexpensive preparation indicates the feasibility of their implementation in strategies for treating waste liquid. As such, it is anticipated that Zr/Fe-MOFs/GO composites will be widely applied in wastewater purification.

## 1. Introduction

With emerging environmental concerns related to clean energy and pollution, environmental remediation has positioned itself at the forefront of conversation and become a prominent area of research. Over the past few decades, due to accelerated urbanization and excessive population growth, copious amounts of organic pollutants have been discharged into environment [1–4]. Environmental pollution has aroused world-wide concern, with particular urgency directed towards water pollution, as the seriousness of this insidious problem is increasing steadily [5, 6]. Although antibiotics have greatly improved the quality and duration of human life, people who drink unclean water containing antibiotic-resistant bacteria may result in a serious consequence of incurable superbugs. In addition, the wastewater discharged from textiles, leather, paper, printing, dyes, plastics, electroplating, and steel manufacturing factories contains a large amount of heavy metals and organic dyes, causing extreme damage to the

Technology Talents Growth Project in the form of an award to QR [KY [2019] 149].

environment and various ecosystems [7, 8]. A long-lasting exposure of organic dyes can cause skin irritation and even cancer or genetic mutations. Substantial effort has been devoted to developing advanced materials to improve the performance of water purification technology.

Over the past 20 years, MOFs have emerged as promising porous materials and attracted increasing attention of researchers in the fields of energy storage [9, 10], adsorption and separation [11–13], catalysis [14, 15], drug delivery [16, 17], carbon dioxide capture [18, 19], chemical sensing [20], antibiotic [21, 22] and others [23–27]. In this regard, photocatalytic reduction methods have demonstrated high selectivity for the pollutant of interest with minimal damage to the ecosystems and are generally inexpensive.

Since organic dyes display varied toxicity and are resistant to photodecomposition and oxidation, they pose serious threats to water quality, and are difficult to purge from the environment. Previously developed technologies rely on physical, chemical, and biological methods such as the use of activated carbon, alginate, and related techniques. Since the inception of MOFs, the number of reports detailing their implementation in water treatment, including the degradation of hexavalent chromium ions and the treatment of organic dyes in wastewater, has increased steadily. Liang and others [28, 29] prepared meth-68(In)-NH$_2$ (40 mg) via solvothermal synthesis with 2-aminoterephthalic acid and indium nitrate in DMF solvent. To test the efficacy of chromium salt degradation, this solution was added to a 20 mg/L Cr(VI) solution (40 mL), followed by addition of H$_2$SO$_4$ and NaOH to adjust the pH, and the system was irradiated with a Xe lamp for 180 min. The highest degradation rate in the ethanol system was 97%, which was 2.25 times and 2.1 times that of ammonium oxalate and ammonium formate, respectively. Wang et al. [30] prepared CMTi by compounding MIL-125(Ti), through the reaction of g-C$_3$N$_4$ with terephthalic acid and tetra-tert-butyl titanate via solvothermal synthesis at 150˚C for 48 h, for the degradation of rhodamine B. A significant degradation (92.5%) of rhodamine was observed after irradiation for 60 min with a 300 W Xe lamp. This highly efficient degradation process is attributed to the adsorption of the substance itself and the π-π interaction between the adsorbent and the substance. Enamul Haque et al. [31] prepared MOF-235 by reacting terephthalic acid with FeCl$_3$·6H$_2$O in DMF solvent for 24 h, for the degradation of methyl orange and methylene blue. The kinetics of adsorption was modeled and estimated to 477 mg/g for methyl orange and 187 mg/g for methylene blue.

For most mono-metal MOFs, the active sites of metal ions for organic ligands are not prominently enough, and the preparation of metal-organic framework materials using bimetallic ions is conducive to the synergistic effect between metal ions. In this paper, we report the development of an efficient MOFs-based adsorbent and its application in removal of two classes of common environmental pollutants. Zirconium has excellent corrosion resistance to a variety of acids, bases and salts. As metal ions of MOFs, Zr and Fe can play a synergistic role in removal of organic pollutants. A key design feature is the use of a bimetallic framework of Zr and Fe, which effectively creates more active sites and results in highly efficient removal of organic pollutants. The composite materials are prepared using solvothermal synthesis with GO, H$_2$BDC, zirconium acetate, and ferrous sulfate heptahydrate as the principle components, and demonstrate effective for removal of tetracycline hydrochloride and orange II.

## 2. Experimental materials and methods

### 2.1 Raw materials

The ligands terephthalic acid (H$_2$BDC, 98%), ferrous sulfate heptahydrate, zirconium acetate, and tetracycline hydrochloride Orange II were purchased from Aladdin Biological Technology Co. LTD (Shanghai, China). Graphene Oxide (GO) was purchased from Beike New Material Technology Co. LTD (Beijing, China).

## 2.2 Preparation of GO/MOFs

The compound was synthesized via hydrothermal synthesis. Dimethylformamide (10 mL) was added to a beaker containing terephthalic acid (3.3215 g) and stirred for 30 min with a magnetic stir bar. Ferrous sulfate heptahydrate (2.8133 g) and zirconium acetate (2.6 mL) were dissolved in distilled water. The prepared solutions were then transferred to a 50 mL reactor and reacted at 120°C for 10 h. Finally, the reaction mixture was filtered and washed thoroughly with ethanol and distilled water. The Zr/Fe-MOFs/GO was dried at 80°C for 12 h.

The sample was analyzed using infrared spectroscopy (IR) with KBr pellets and exhibted key signals at 2000–400 cm$^{-1}$. Further analyses were performed with an XRD diffractometer (D-5000XRD, Liaoning Dandong Tongda Science and Technology Co. Ltd., China) under the conditions of 30 kv and 20 mA with a 2θ scan range of 5–80, a field emission scanning electron microscope (FESEM, JSM-6700F, Japan), and a UV spectrometer (UV-2550, Shimadzu, Japan).

## 2.3 Removal of organic contaminant

Tetracycline hydrochloride and Orange II were chosen as model contaminants to evaluate the degradation ability of Zr/Fe-MOFs/GO. Solutions of varying concentrations (20 ppm, 30 ppm, 40 ppm, and 50 ppm) of tetracycline hydrochloride and Orange II were prepared. The Zr/Fe-MOF (50 mg) was added to each solution and stirred under natural visible light. The concentration of each analyte was measured every hour using UV-visible spectroscopy (tetracycline hydrochloride, 360 nm, and orange II, 485 nm) [32] to determine the rate of degradation as follows:

$$q_e = \frac{(C_0 - C_e)V}{m} \tag{1}$$

$C_e$, $C_0$, V and m are the equilibrium concentrations of the solution (ppm), the initial concentrations of the solution (ppm), the volume of the solution (L), and the mass of the GO/MOFs, respectively.

## 3. Results and discussion

### 3.1 Structural characterization

The IR spectrum (Fig 1) of the GO/MOFs showed strong absorption peaks at 1402 cm$^{-1}$ and 1677 cm$^{-1}$. The presence of an intense carbonyl peak at ca. 1710 cm$^{-1}$ is attributed to the equivalency of the two carbonyl groups in the carboxylate-coordinated metal ion, in which electron clouds tend to be delocalized due to the conjugated $\pi$ bond of $\pi_3^5$. Instead, relatively strong absorption peaks at 1610–1550 cm$^{-1}$ and 1420–1300 cm$^{-1}$ were observed. These two peaks provide diagnostic signals for evaluating the reactivity of the carboxylic acids with the metal salts.

After confirming the presence of the desired MOF by XRD (Fig 2), the remaining metal salt and terephthalic acid can be easily removed with aqueous and organic washes, respectively. Therefore, it is clear that the product generated is a new substance. As evidenced by SEM images (Fig 3), the composite formed consists of GO with layered and sheet-like structures containing electrons and MOFs, which tend to have different morphological structures, leading to significantly different structural features.

Fig 4 shows that the surface area of GO/MOFs was 1.5061 m$^2$/g. The single point surface area at P/Po = 0.249224654 was 8.9549 m$^2$/g. The average particle size was 85.2394 nm, and the t-Plot micropore volume was 0.004155 cm$^3$/g, indicating a mesoporous material.

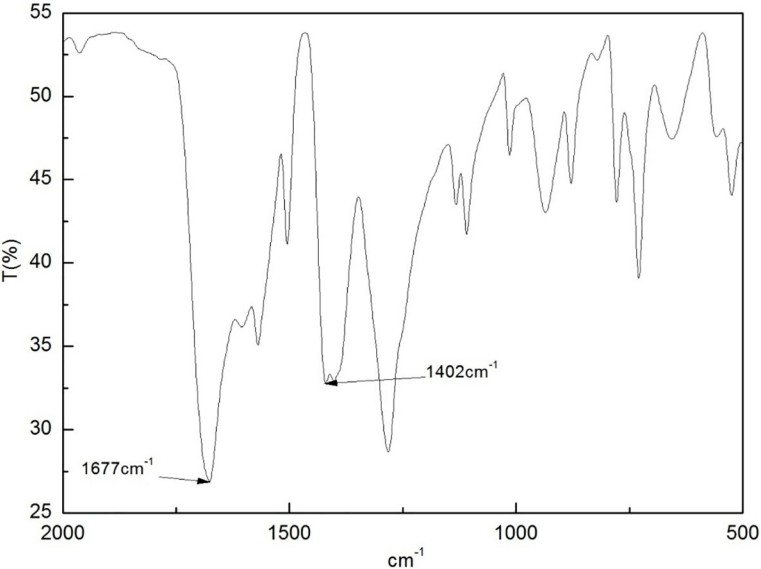

**Fig 1. IR of MOFs.**

## 3.2 Removal of organic pollutant by MOFs

The concentration of pollutants is a key factor that affects their adsorption in wastewater treatment. By implementing key structural changes in the MOF framework, the rate of pollutant removal was markedly increased. Competition of tetracycline hydrochloride and orange II for active sites on the MOF has a profound impact on the rate of pollutant degradation. The decomposition products can also compete for binding, which further indicates the necessity for high selectivity of the pollutant of interest [33]. In addition, when the concentrations of tetracycline hydrochloride and orange II are high, photons are not able to effectively penetrate

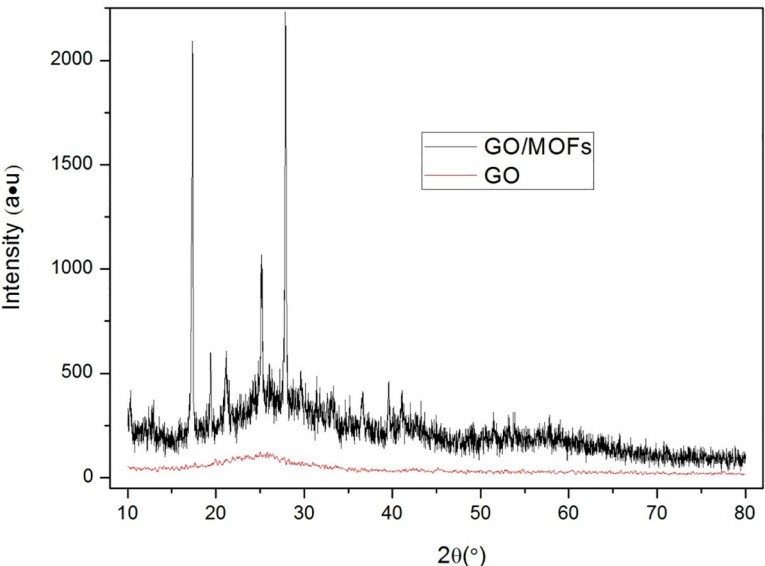

**Fig 2. XRD of GO/MOFs.**

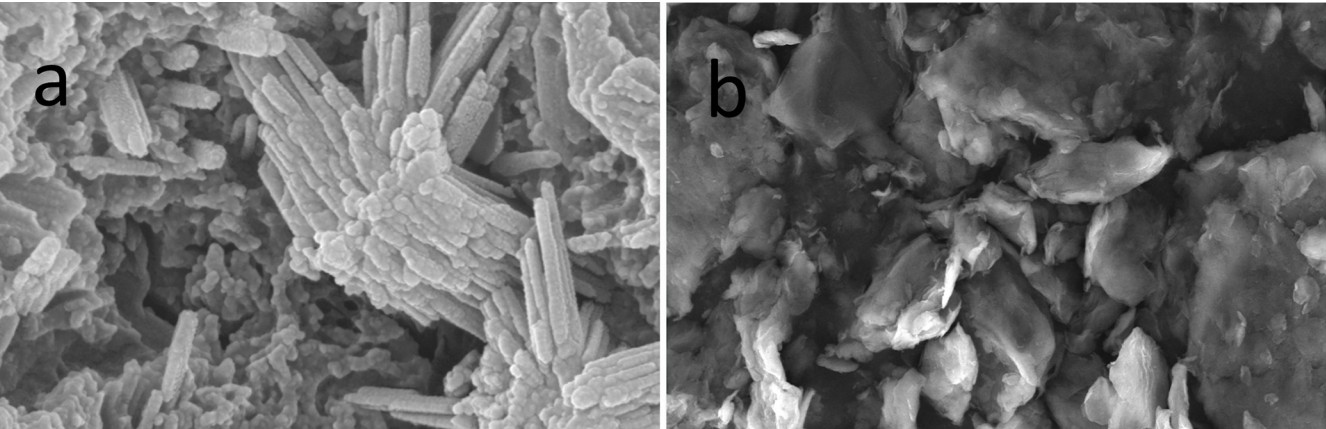

**Fig 3.** SEM of GO/MOFs (a) and GO (b).

the solution, which in turn slows the rate of degradation. Despite these challenges, the Zr/Fe-MOFs/GO designed here demonstrates high rates of pollutant removal for tetracycline hydrochloride and Orange II at relatively low concentrations.

In order to investigate the efficacy of MOFs designed here in removing organic pollutants, we studied the removal of tetracycline hydrochloride and orange II at varying concentrations. Zr/Fe-MOFs/GO (50 mg) was mixed with 100 mL of 10 mg/L, 20 mg/L, 30 mg/L, or 50 mg/L tetracycline hydrochloride solution and 20 mg/L, 30 mg/L, 50 mg/L, or 60 mg/L orange II solution. The results revealed that the lower the concentration, the higher the removal rate of the pollutants.

Upon adsorption of pollutants, the Zr/Fe-MOFs/GO tends to precipitate thus effectively reducing the concentration of active sites in solution, and slowing the rate of degradation [34, 35]. To determine the reusability of the Zr/Fe-MOFs/GO, the used Zr/Fe-MOFs/GO was

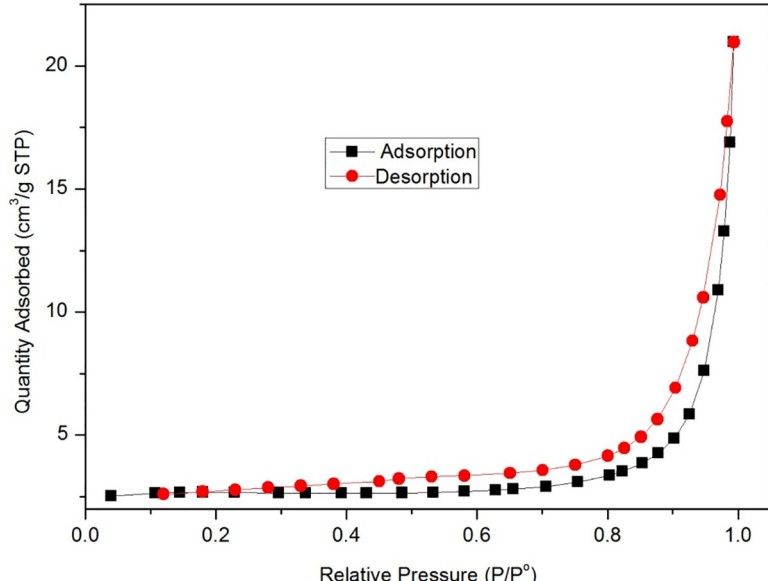

**Fig 4. N2 adsorption-desorption isotherms of GO/MOFs.**

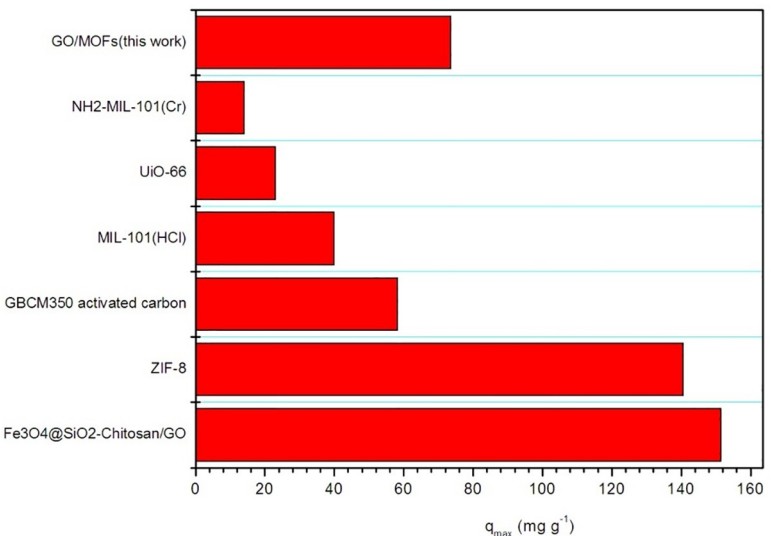

**Fig 5. Comparison of the GO/MOFs adsorbent with other materials on TC adsorption.**

washed with water, dried, and reused. The results showed that after three cycles, the decrease in activity was only 39% and 31%, which indicates a reasonable robustness, and the possibility for reusability. The results of TC removal by Zr/Fe-MOFs/GO and GBCM350 activated carbon [36], $Fe_3O_4@SiO_2$-chitosan/GO [37], ZIF-8 [38], $NH_2$-MIL-101(Cr) [39], MIL-101(HCl) [40], and Uio-66 [41] were compared, and Fig 5 shows the superior activity of Zr/Fe-MOFs/GO relative to other adsorbents.

In order to fully understand the removal of pollutants using the GO/MOFs, the kinetics of tetracycline and orange II degradation was examined. In accordance with reports in the literature, removal of contaminants was described by the following kinetic equations [42–44]:

the Pseudo-first-order kinetic model:

$$\ln\frac{C_t}{C_0} = k_1 t \tag{2}$$

the Pseudo-second-order kinetic model:

$$\frac{t}{q_t} = \frac{t}{q_e} + \frac{1}{k_2 q_e^2} \tag{3}$$

In these equations, $C_t$, $C_0$, $k_1$, $k_2$ and t are the concentration of tetracycline at time t, the initial concentration of tetracycline and orange II, the reaction rate constant ($min^{-1}$), and the reaction time (min). $q_t$ and $q_e$ represent the amounts ($mg \cdot g^{-1}$) of the adsorbents at time t and equilibrium, respectively. The adsorption results of GO /MOFs on tetracycline hydrochloride and orange II are shown in Figs 6 and 7 and Table 1. The pseudo-first order model and pseudo-second order model both appropriately describe the removal of tetracycline hydrochloride.

In order to study the removal of organic dyes by GO/MOFs, Orange II was chosen as the model pollutant. The kinetic results are shown in Figs 8 and 9 and Table 2, and its degradation is well described by pseudo-second-order kinetics.

Based on the above results, the mechanism of tetracycline hydrochloride and orange II removal by GO/MOFs composite material is as follows. The specific surface area and pores of

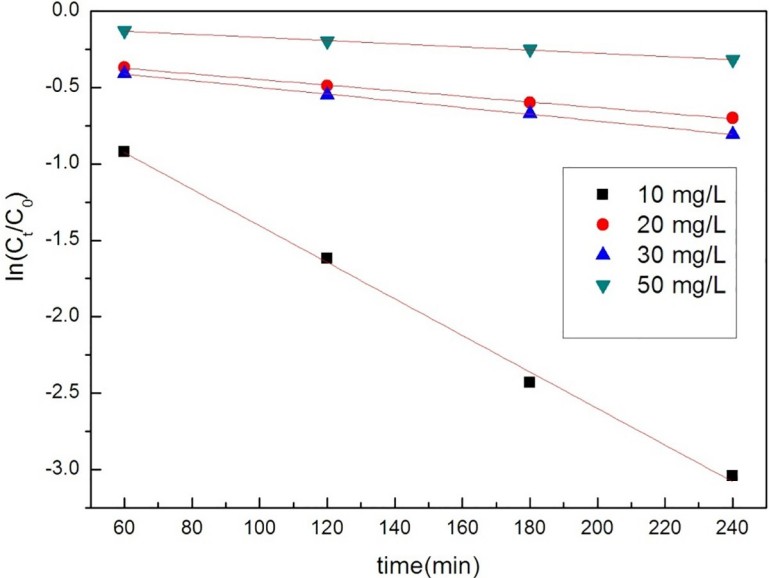

**Fig 6. The Pseudo-first-order kinetic model of GO/MOFs on TC.**

the GO/MOFs composite material facilitate the diffusion of tetracycline hydrochloride and orange II from the solution to the surface and pores of the GO/MOFs [45–47]. This is driven by favorable interactions between the analytes and the GO/MOFs, in which polar groups, such as hydroxyl and amino groups, engage in hydrogen bonding with the hydrophilic groups on the GO/MOFs. In addition, π-systems present in both the analytes and the GO/MOFs also facilitate favorable π-π interactions [48–51]. As a result, the surface electronegativity of the GO/MOFs composite changes slightly, which further affects the electrostatic interaction between the GO/MOFs and tetracycline hydrochloride or orange II. Taken together, these key

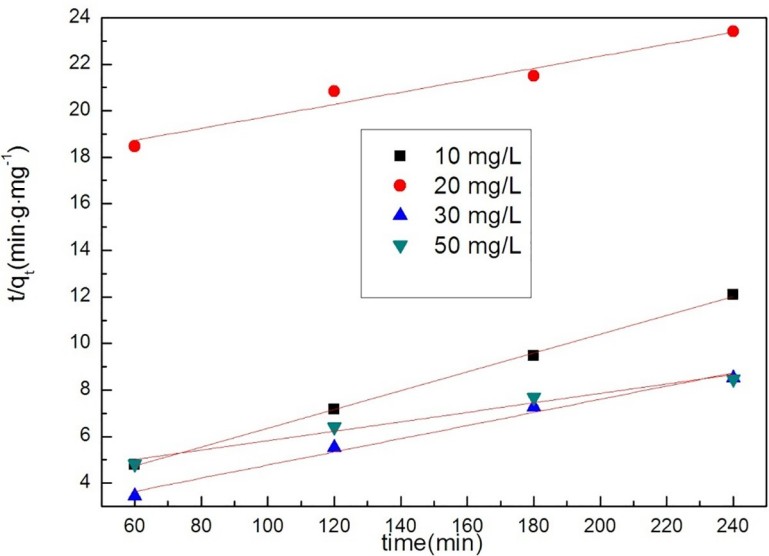

**Fig 7. The Pseudo-second-order kinetic model of GO/MOFs on TC.**

**Table 1. Parameters of the process of tetracycline hydrochloride by GO/MOFs.**

| Concentration | pseudo-first order | | pseudo-second order | |
|---|---|---|---|---|
| | K | $R^2$ | K | $R^2$ |
| 50 mg/L | 0.00103 | 0.99378 | 0.02033 | 0.96709 |
| 30 mg/L | -0.0022 | 0.99862 | 0.02826 | 0.98143 |
| 20 mg/L | -0.00183 | 0.99752 | 0.0258 | 0.94197 |
| 10mg/L | -0.01195 | 0.99602 | 0.04032 | 0.96709 |

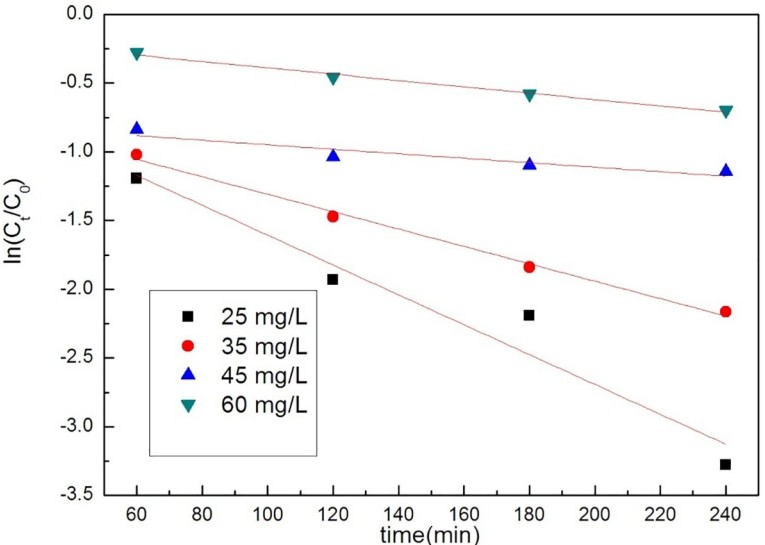

**Fig 8. The Pseudo-first-order kinetic model of GO/MOFs on Orange II.**

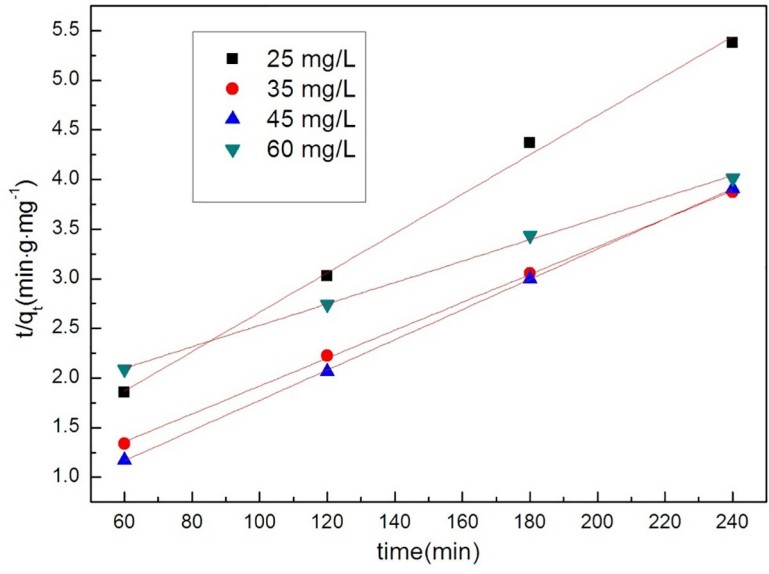

**Fig 9. The Pseudo-second-order kinetic model of GO/MOFs on orange II.**

**Table 2. Parameters of the process of orange II by GO/MOFs.**

| Concentration | pseudo-first order | | pseudo-second order | |
|---|---|---|---|---|
| | K | $R^2$ | K | $R^2$ |
| 60 mg/L | 0.00232 | 0.98146 | 0.01078 | 0.99798 |
| 45 mg/L | -0.00164 | 0.82498 | 0.01523 | 0.99989 |
| 35 mg/L | -0.00633 | 0.99139 | 0.01405 | 0.99949 |
| 25 mg/L | -0.01086 | 0.9221 | 0.01983 | 0.99586 |

interactions including hydrogen bonding, π-π interactions, and electrostatic attraction all contribute to the high efficacy of GO/MOFs mediated degradation of tetracycline hydrochloride and orange II.

## 4. Conclusion

GO/MOFs have been successfully prepared by solvothermal synthesis and characterized with IR, XRD and SEM. The MOF has demonstrated excellent efficiency in degradation of tetracycline hydrochloride and orange II under natural light irradiation which has important implications in water remediation technology. The experimental findings of pseudo-first-order decay and pseudo-second-order decay are in excellent agreement with similar systems reported in the literature. In addition to achieving an excellent pollutant removal strategy, the results presented herein also explore alternative avenues to prepare metal organic frameworks in a highly efficient manner. Therefore, it is anticipated that GO/MOFs composites will find widespread application for the removal of organic pollutants in a variety of contexts in water purification area.

## Supporting information

**S1 File.**
(RAR)

## Author Contributions

**Data curation:** Zhao Liang.

**Formal analysis:** Hongliang Chen.

**Funding acquisition:** Qinhui Ren.

**Investigation:** Huan Zhang, Bo Ding, Mengjie Yu.

**Software:** Lili Yang.

**Writing – original draft:** Fuhua Wei.

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
