## [Decision Letter · Decision Letter 0]

1 Apr 2021

PONE-D-21-03199

Removal of organic contaminants from wastewater with GO/MOFs composites

PLOS ONE

Dear Dr. Wei,

Thank you for submitting your manuscript to PLOS ONE. After careful consideration, we feel that it has merit but does not fully meet PLOS ONE’s publication criteria as it currently stands. Therefore, we invite you to submit a revised version of the manuscript that CAREFULLY addresses the points raised during the review process.

We look forward to receiving your revised manuscript.

Kind regards,

Yogendra Kumar Mishra, Ph. D.

Academic Editor

PLOS ONE

Journal Requirements:

2a) If there are ethical or legal restrictions on sharing a de-identified data set, please explain them in detail (e.g., data contain potentially sensitive information, data are owned by a third-party organization, etc.) and who has imposed them (e.g., an ethics committee). Please also provide contact information for a data access committee, ethics committee, or other institutional body to which data requests may be sent.

2b) If there are no restrictions, please upload the minimal anonymized data set necessary to replicate your study findings as either Supporting Information files or to a stable, public repository and provide us with the relevant URLs, DOIs, or accession numbers. For a list of acceptable repositories, please see http://journals.plos.org/plosone/s/data-availability#loc-recommended-repositories.

3. Please ensure that you refer to Figure 3 in your text as, if accepted, production will need this reference to link the reader to the figure.

Reviewers' comments:

Reviewer's Responses to Questions

**Comments to the Author**

1. Is the manuscript technically sound, and do the data support the conclusions?

Reviewer #1: Yes

Reviewer #2: Yes

2. Has the statistical analysis been performed appropriately and rigorously? 

Reviewer #1: Yes

Reviewer #2: No

3. Have the authors made all data underlying the findings in their manuscript fully available?

Reviewer #1: Yes

Reviewer #2: Yes

4. Is the manuscript presented in an intelligible fashion and written in standard English?

Reviewer #1: Yes

Reviewer #2: Yes

5. Review Comments to the Author

Reviewer #1: In this manuscript, synthesis, characterization and organic contaminants removal performance of GO/MOFs composite is described. The methodology and applications presented in the manuscript are interesting and will appeal to a broader readership of Plos One. The manuscript is recommended for publication subjected to addressing following comments:

1. GO/MOFs first explained then abbreviation should be used.

2. The introduction section needs to be modified as the discussion appears to be more scattered in nature.

3. To gain the porous parameters (e. g. the specific surface area, total pore volume, the surface area and pore volume of the microspores), N2 adsorption-desorption experiments for GO/MOFs are suggested to supply.

4. Proper characterization of GO like XRD, SEM and Raman should be included.

5. The dye absorption behavior, suggesting authors use commercial activated carbon as the benchmark; and evaluate the absorption efficiency of these GO/MOFs composite.

6. The cycle performance of the adsorbent is suggested to be checked.

7. The English should be carefully polished. Also, many format problems should be addressed.

8. The quality of ALL OF FIGURES should be further decorated. The following literature should be consulted and employed: Environmental Science and Pollution Research, 2020, 27, 32874–32887; ACS Applied Materials and Interfaces, 2019, 11, 18165-18177; ACS Applied Materials and Interfaces, 2019, 11, 43949-43963.

Reviewer #2: The manuscript entitiled “Removal of organic contaminantsfrom wastewater with GO/MOFscomposites” submitted by Fuhua Wei et al is an nice submission regarding Environmental sciences. The idea , concept and the methodology of the manuscript is novel as well as well defined however the manuscript needs to be corrected before acceptance. The comments are :

1. Please use full form of abbreviation when using first time in the manuscript. What is MOF means here?

2. Line 35-39, meaning is unclear with grammatical mistakes. Rephrase it.

3. Author has mentioned many times about the toxicity of the previous materials in introduction. They need to be clear with statement about the biological toxicity with proper citations. In general, toxicity applied to the biological toxicity of materials, metals or dyes to the living cells present in environmental sources. Author should focus also in this issue in the introduction. Some suggested citations are mentioned in the last comment.

4. Introduction should also mention, what is the need of the new material and what is the hypothesis of synthesis of this new material.

5. Why Zr and Fe? Any specific properties which led to better environmental application? Mention in introduction.

6. It will be better if, author can present the mechanism mentioned in the result and discussion section through a schematic diagram.

7. Lastly authors are suggested to cite some recent articles about the metal toxicity in environment. Some of the suggested one are: DOI: 10.1039/C7RA05943D, https://doi.org/10.1093/toxsci/kfx204, https://doi.org/10.1016/j.msec.2018.07.037,https://doi.org/10.1080/21691401.2018.1503598 , DOI: 10.1016/j.msec.2021.111888 , DOI: 10.1016/j.bioorg.2020.104535 , DOI: 10.2217/nnm-2020-0138, DOI: 10.1016/j.mtchem.2020.100299; DOI: 10.1016/j.mtchem.2020.100345; https://doi.org/10.1021/acsomega.7b01522, https://doi.org/10.1016/j.jcis.2018.07.020 , DOI: 10.1016/j.envpol.2020.115482

6. PLOS authors have the option to publish the peer review history of their article (what does this mean?). If published, this will include your full peer review and any attached files.

Reviewer #1: No

Reviewer #2: No

---

## [Author Response · Author response to Decision Letter 0]

6 May 2021

Dear Editors and Reviewers:

 Thank you for your letter and for the reviewers’ comments concerning our manuscript entitled “Removal of organic contaminants from wastewater with GO/MOFs composites”. Those comments are all valuable and very helpful for revising and improving our paper. We have studied comments carefully and tried our best to revise the manuscript. Revised portion are marked in red in the paper. The point to point responds to the reviewer’s comments are listed as following:

 Responds to the reviewer’s comments:

Reviewer#1

Comments to the Author

In this manuscript, synthesis, characterization and organic contaminants removal performance of GO/MOFs composite is described. The methodology and applications presented in the manuscript are interesting and will appeal to a broader readership of Plos One. The manuscript is recommended for publication subjected to addressing following comments:

Comment 1: GO/MOFs first explained then abbreviation should be used.

Response to Comment 1: Thank you for your careful reading of our manuscript.According to your comment, we have added the interpretation of GO/MOFs in the revised paper. Revised portion are marked in red in the paper. (Page 1,lines 10)

Comment 2: The introduction section needs to be modified as the discussion appears to be more scattered in nature.

Response to Comment 2: Thank you for your valuable advice. According to your comment, the introduction section have modified in the revised paper. (Page 2-3) 

Comment 3: To gain the porous parameters (e. g. the specific surface area, total pore volume, the surface area and pore volume of the microspores), N2 adsorption-desorption experiments for GO/MOFs are suggested to supply.

Response to Comment 3: Thank you for your careful reading of our manuscript. According to your comment, We have added the Figure 4 , the surface area and pore volume of the microspores in the revised paper. (Page 7,lines145-148;Page 8,Fig.4)

Comment 4: Proper characterization of GO like XRD, SEM and Raman should be included.

Response to Comment 4:Thank you for your valuable suggestion. According to your comment, We have added the characterization of GO in the revised paper. (Page 6,Fig.2;page 7,Fig.3(b))

Comment 5: The dye absorption behavior, suggesting authors use commercial activated carbon as the benchmark; and evaluate the absorption efficiency of these GO/MOFs composite.

Response to Comment 5: Thank you for your valuable advice. We have purchased the activated carbon, but the manufacturer has not delivered the product for a long time. We are very sorry about it.The data will be added to future research.

Comment 6. The cycle performance of the adsorbent is suggested to be checked.

Response to Comment 6: Thank you for your careful reading of our manuscript. We are sorry about the mistakes. According to your comment, The cycle performance of the adsorbent have corrected in the revised paper. Revised portion are marked in red in the paper.(Page 9,lines 172-173)

Comment 7. The English should be carefully polished. Also, many format problems should be addressed.

Response to Comment 7: Thank you for your careful reading of our manuscript. We are sorry about the mistakes. According to your comment, the revised paper have checked by a native English speaker because there are some grammatical and spelling errors that have to be corrected. Revised portion are marked in red in the paper.

Comment 8. The quality of ALL OF FIGURES should be further decorated. The following literature should be consulted and employed: Environmental Science and Pollution Research, 2020, 27, 32874–32887; ACS Applied Materials and Interfaces, 2019, 11, 18165-18177; ACS Applied Materials and Interfaces, 2019, 11, 43949-43963.

Response to Comment 8: Thank you for your valuable advice. According to your comment, we have added the reference [7]、[2]、[12]in the revised paper. Revised portion are marked in red in the paper. 

Reviewer#2:

The manuscript entitiled “Removal of organic contaminantsfrom wastewater with GO/MOFscomposites” submitted by Fuhua Wei et al is an nice submission regarding Environmental sciences. The idea , concept and the methodology of the manuscript is novel as well as well defined however the manuscript needs to be corrected before acceptance. The comments are :

Comment 1: Please use full form of abbreviation when using first time in the manuscript. What is MOF means here?

Response to Comment 1: Thank you for your careful reading of our manuscript .We are sorry about the mistakes. According to your comment, we have added the interpretation of GO/MOFs in the revised paper. Revised portion are marked in red in the paper. (Page 1,lines 10) 

Comment 2: Line 35-39, meaning is unclear with grammatical mistakes. Rephrase it.

Response to Comment 2: Thank you for your careful reading of our manuscript. We are sorry about the mistakes. According to your comment, the revised paper have checked by a native English speaker because there are some grammatical and spelling errors that have to be corrected. Revised portion are marked in red in the paper. (Page 2,lines 35-38)

Comment 3: Author has mentioned many times about the toxicity of the previous materials in introduction. They need to be clear with statement about the biological toxicity with proper citations. In general, toxicity applied to the biological toxicity of materials, metals or dyes to the living cells present in environmental sources. Author should focus also in this issue in the introduction. Some suggested citations are mentioned in the last comment.

Response to Comment 3: Thank you for your valuable advice. According to your comment, we have added the reference [3]、[5]、[6]、[21]、[22]、[23]、[24]in the revised paper. Revised portion are marked in red in the paper. 

Comment 4: Introduction should also mention, what is the need of the new material and what is the hypothesis of synthesis of this new material.

Response to Comment 4: Thank you for your careful reading of our manuscript. According to your comment, we have added the content in the revised paper. Revised portion are marked in red in the paper. (Page 3,lines 76-81) 

Comment 5: Why Zr and Fe? Any specific properties which led to better environmental application? Mention in introduction.

Response to Comment 5: Thank you for your careful reading of our manuscript. According to your comment, we have added the content in the revised paper. Revised portion are marked in red in the paper. (Page 3,lines 82-85)

Comment 6. It will be better if, author can present the mechanism mentioned in the result and discussion section through a schematic diagram.

Response to Comment 6: Thank you for your careful reading of our manuscript. According to your comment, Due to time problems, the schematic diagram could not be provided in time. We are sorry about it. The schematic diagram will be added to future research.

Comment 7. Lastly authors are suggested to cite some recent articles about the metal toxicity in environment. Some of the suggested one are: DOI: 10.1039/C7RA05943D, https://doi.org/10.1093/toxsci/kfx204, https://doi.org/10.1016/j.msec.2018.07.037,https://doi.org/10.1080/21691401.2018.1503598 , DOI: 10.1016/j.msec.2021.111888 , DOI: 10.1016/j.bioorg.2020.104535 , DOI: 10.2217/nnm-2020-0138, DOI: 10.1016/j.mtchem.2020.100299; DOI: 10.1016/j.mtchem.2020.100345; https://doi.org/10.1021/acsomega.7b01522, https://doi.org/10.1016/j.jcis.2018.07.020 , DOI: 10.1016/j.envpol.2020.115482

Response to Comment 7: Thank you for your valuable advice. According to your comment, we have added the reference [3]、[5]、[6]、[21]、[22]、[23]、[24]in the revised paper. Revised portion are marked in red in the paper.

---

## [Decision Letter · Decision Letter 1]

7 Jun 2021

Removal of organic contaminants from wastewater with GO/MOFs composites

PONE-D-21-03199R1

Dear Dr. Wei,

We’re pleased to inform you that your manuscript has been judged scientifically suitable for publication and will be formally accepted for publication once it meets all outstanding technical requirements.

Kind regards,

Yogendra Kumar Mishra, Ph. D.

Academic Editor

PLOS ONE

Additional Editor Comments (optional):

Reviewers' comments:

Reviewer's Responses to Questions

**Comments to the Author**

1. If the authors have adequately addressed your comments raised in a previous round of review and you feel that this manuscript is now acceptable for publication, you may indicate that here to bypass the “Comments to the Author” section, enter your conflict of interest statement in the “Confidential to Editor” section, and submit your "Accept" recommendation.

Reviewer #1: All comments have been addressed

Reviewer #2: All comments have been addressed

2. Is the manuscript technically sound, and do the data support the conclusions?

Reviewer #1: Yes

Reviewer #2: Yes

3. Has the statistical analysis been performed appropriately and rigorously? 

Reviewer #1: Yes

Reviewer #2: Yes

4. Have the authors made all data underlying the findings in their manuscript fully available?

Reviewer #1: Yes

Reviewer #2: Yes

5. Is the manuscript presented in an intelligible fashion and written in standard English?

Reviewer #1: Yes

Reviewer #2: Yes

6. Review Comments to the Author

Reviewer #1: (No Response)

Reviewer #2: the authors have addressed all the comments and made the revision properly.The manuscript can be recommended for acceptance.

7. PLOS authors have the option to publish the peer review history of their article (what does this mean?). If published, this will include your full peer review and any attached files.

Reviewer #1: No

Reviewer #2: No

---

## [Editor Report · Acceptance letter]

21 Jun 2021

PONE-D-21-03199R1 

Removal of organic contaminants from wastewater with GO/MOFs composites 

Dear Dr. Wei:

I'm pleased to inform you that your manuscript has been deemed suitable for publication in PLOS ONE. Congratulations! Your manuscript is now with our production department. 

Kind regards, 

on behalf of

Professor Yogendra Kumar Mishra 

Academic Editor

PLOS ONE